# Towards Sustainable Digital Agriculture for Smallholder Farmers: A Systematic Literature Review

**Nametshego Gumbi** , **Lucas Gumbi \*** and **Hossana Twinomurinzi \***

Centre for Applied Data Science, University of Johannesburg, Auckland Park,
P.O. Box 524, Johannesburg 2006, South Africa; nametshego.m@gmail.com
\* Correspondence: lnhlanhla@yahoo.com (L.G.); hossanat@uj.ac.za (H.T.); Tel.: +27-64-521-8527 (H.T.)

**Abstract:** Smallholder farmers are key contributors to food security globally, and more so in developing countries. Despite their critical role in food security, smallholder farmers are highly constrained by specific contextual challenges such as climate change, productivity, cost of production, credit access, and financial resources constraints that impact their sustenance, sustainability, and growth. Digital agriculture has emerged as a viable solution to addressing smallholder farmers' contextual challenges, with many digital solutions already existing and developed to serve the agriculture sector. However, many smallholder farmers are beyond the reach of these digital solutions due to underdeveloped or nonexistent digital ecosystems. This paper reports on a systematic review conducted to examine the research that has been undertaken regarding digital agriculture ecosystems in relation to smallholder farmers and to identify challenges, usage, benefits, access, and uptake of the systems. The key findings reveal very limited research directed at digital literacy or skills, affordability, and business model innovation. Most of the challenges concern digital infrastructure, affordability, and digital literacy or skills. The findings also reveal that although digital agriculture is still a nascent concept to smallholder farmers, there are a few early adopters who access information mainly related to agriculture, selling, and marketing. There is, nonetheless, a lack of understanding of the value of digital agriculture systems. The study develops a research agenda that could facilitate digital transformation for smallholder farmers.

**Keywords:** digital agriculture; digital platforms; digital literacy; digital skills; 4IR; affordability; smallholder farmers; systematic literature review

## 1. Introduction

The agriculture sector plays a key role in socioeconomic development globally. The sector is critical for ensuring food security as well as eliminating poverty (Sustainable Development Goal 1) and hunger (Sustainable Development Goal 2) [1,2]. The sector also plays an important role in improving employment, income, and standards of living for the poor [1,2].

Food security has been identified as a high priority in the developing regions of the world. Population growth coupled with the increased intensity of environmental events such as floods and droughts, along with higher food prices and income inequalities, often pose a threat to food access and availability for poor households in the developing regions of the world [2].

Smallholder farmers in developing countries play a critical role worldwide in ensuring food security [3,4]. While there is no standard or universally accepted definition of smallholder farmers, they are characterized as those farmers that cultivate small areas of land (usually less than 10 ha and often less than 2 ha), use family labor, and depend on their farms as their main source of both food security and income generation [5]. There are approximately 500 million smallholder farmers globally who provide an estimated 70–80% of the food produced in Asia and sub-Saharan Africa [3,4]. This makes smallholder

farmers critically important and indispensable for food security in sub-Saharan Africa and developing countries in general.

Agriculture is an important asset for livelihood in poor rural communities in developing and low-income countries [6–8]. Smallholder farmers in developing and low-income countries are the primary source of food security, income, employment, poverty alleviation, and the improvement of livelihoods in rural areas. However, smallholder farmers in developing and low-income countries are highly constrained by specific contextual challenges such as climate change, credit access, failures of agricultural policies, constraints of food safety standards, poor infrastructure and business environments, historical background, geographic location, unbalanced land distribution, deficiency of appropriate skills and manpower, and the cost of telecommunication and Internet accessibility [9–11].

Digital agriculture, sometimes referred to as smart agriculture, smart farming, precision agriculture, or agriculture 4.0, has been identified as a viable solution to addressing smallholder farmers' issues [12]. An example is the AgroTech Smartex Digital Platform designed and implemented by the Grameen Foundation and partners with the aim of improving farm business productivity and profitability [12]. However, digital agriculture ecosystems are much more developed in urban areas compared with rural areas [13].

Many digital solutions developed to serve the agriculture sector already exist [14,15]. However, many smallholder farmers are beyond the reach of these digital solutions due to underdeveloped or nonexistent digital ecosystems [13]. Some of the challenges that make these digital solutions unsuitable for smallholder farmers' contexts are the costs of service (affordability and lack of business model innovation), digital skills and digital literacy gaps (impacting the use of digital platforms), a lack of digital technologies (4IR) including smart devices, and digital infrastructure such as connectivity and mobile networking infrastructure for the Internet and broadband connectivity. Figure 1 depicts some of the elements of a digital agriculture ecosystem: namely, the digital platform to access agriculture information, markets, finance, agri-inputs, supply chain management, advisory services, and business intelligence; business model innovation; digital literacy or skills; digital infrastructure; 4IR (artificial intelligence (AI), cloud, big data, the Internet of Things (IoT), and smart and remote sensors); and affordability.

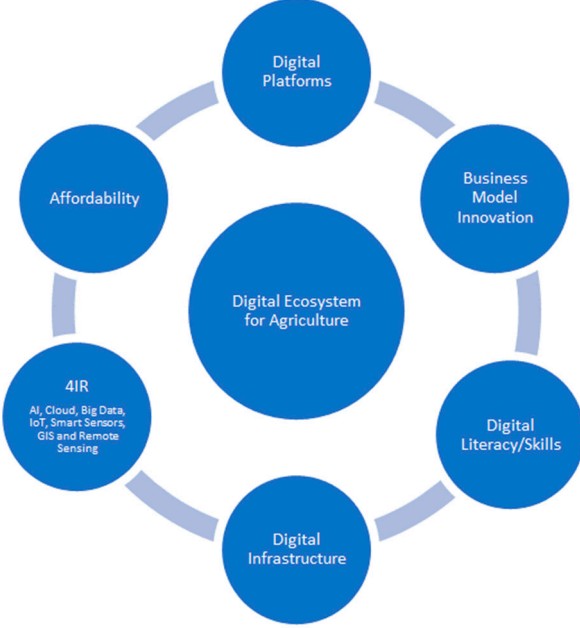

**Figure 1.** Digital agriculture ecosystem.

The aim of this paper was to investigate how digital agriculture ecosystems have been researched regarding smallholder farmers. The investigation was carried out through a

systematic literature review (SLR) and covered the research performed from 2017 to 2022. The SLR sought to find answers to the following research questions (RQs):

**RQ1.** What research has been undertaken in relation to digital ecosystems for small-holder farmers?

**RQ2.** What are the challenges of digital agriculture ecosystems in relation to small-holder farmers?

**RQ3.** How are smallholder farmers using digital solutions in their businesses?

**RQ4.** What are the factors that influence the uptake of digital solutions by smallholder farmers?

**RQ5.** What are the benefits of digital solutions for smallholder farmers?

**RQ6.** What is the level of access to and uptake of digital solutions by smallholder farmers?

## 2. Methodology

To establish a structured approach and a guiding framework for the analysis of the research, the study followed Amui et al.'s (2017) [16] systematic review protocol shown in Figure 2. The protocol is similar to that proposed by Lage Junior and Godinho Filho [17].

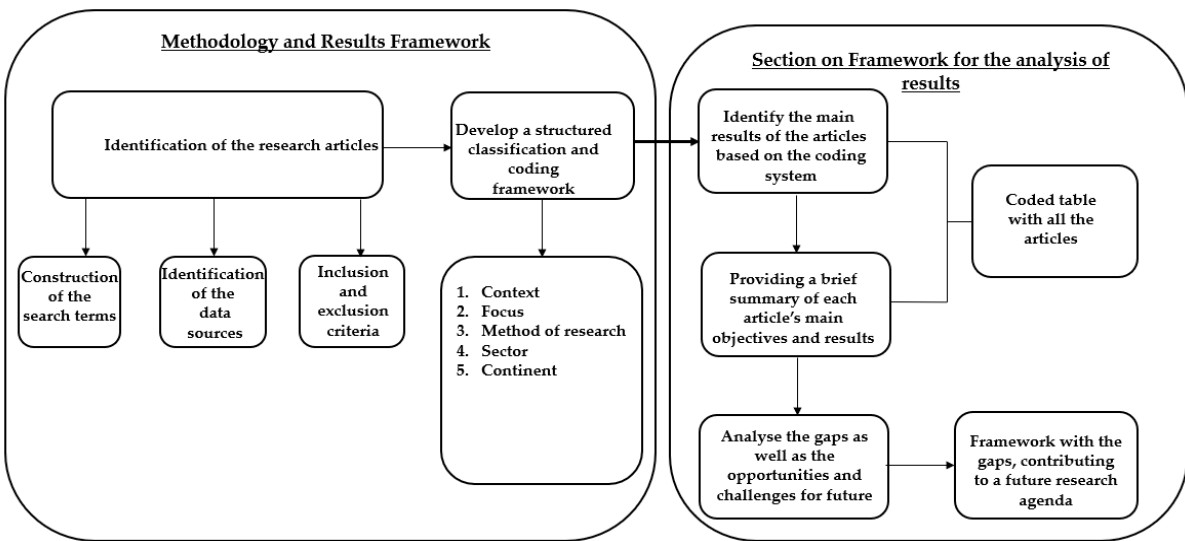

**Figure 2.** A systematic review protocol for the study.

### 2.1. Identification of the Research Articles

The first steps were (1) the construction of the search terms, (2) the identification of the relevant data sources, and (3) the application of inclusion and exclusion criteria as indicated in Figure 2.

#### 2.1.1. Construction of the Search Terms

The search terms were derived from the unit of analysis (smallholder farmers) and technology artifact (digital agriculture ecosystem). The search strings were constructed using the identified unit of analysis (smallholder farmers) and technology artifact (digital agriculture ecosystem). The dimensions (factors) of interest (challenges, access to, uptake, factors, use, and benefits) emerged from the SLR guided by the research questions (RQ1 to RQ6). The full details of the constructed search strings that were used to identify suitable research articles are presented in Appendix A.

#### 2.1.2. Identification of the Data Sources

The study falls under the interdisciplinary field of information systems (IS) and therefore used the following multidisciplinary databases as the data sources: Web of

Science, Academic Search Premier, Science Direct, and Google Scholar. These databases are consistently utilized by many peer-reviewed SLR studies [18–26].

### 2.1.3. Inclusion and Exclusion Criteria

After using the same constructed search terms for all databases, a total of 4837 articles were identified (Figure 3). The first stage of the selection process involved the selection of papers based on the relevancy of the title, abstract, and keywords. This resulted in a total of 173 articles. Thus, in the first stage of sorting, 4664 articles were excluded from the identified articles. The second stage of the selection process was based on the following inclusion criteria: to include journals or conference articles that mentioned the specific terms for the unit of analysis and technology artifact (Appendix A) and the dimensions of interest within their content or title; were written in the English language; were academically peer-reviewed; were published in a six-year period (2017 to 2022); were focused at a firm level and not at an individual level; and were also focused on the agriculture sector or industries. In the second sorting, after removing duplicates in each database separately using the title and abstract, a total of six articles were further excluded from the identified articles.

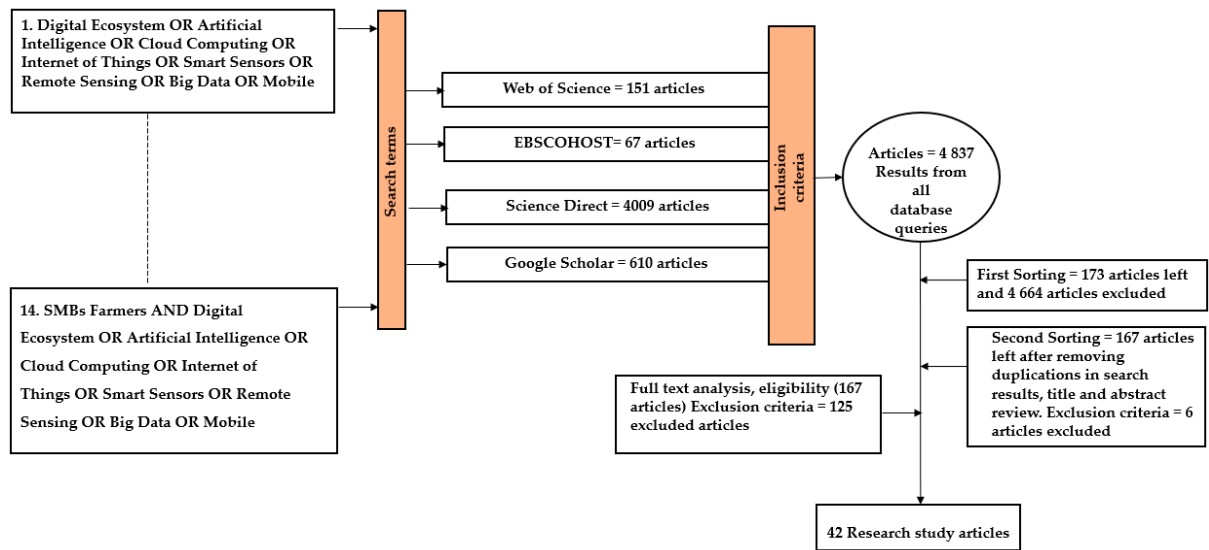

**Figure 3.** A schematic representation of the inclusion and exclusion process.

In the third iteration, exclusion criteria were used to exclude articles that failed the eligibility test because they mentioned only some of the specific search terms but did not solely focus on the research topic, digital ecosystems, nor did they focus on agriculture digital ecosystems; were practitioner-based and not based on any scientific research method; and did not focus on smallholder farmers but merely mentioned smallholder farmers. In the final sorting, 125 articles were excluded based on the analysis of the full text of the articles, and only 42 peer-reviewed journal and conference papers remained, as shown in Figure 3. The primary exclusion criteria were articles that did not mention digital agriculture ecosystem, or digital ecosystem for agriculture, or smallholder farmers.

### 2.2. Coding and Classification Framework

The systematic review was analyzed using the coding and classification framework developed by Amui et al. (2017) [16] based on other works [17,27–30]. The classification was performed using the following categories: context, focus, method, sector, and global locations in the form of origin or continents, as shown in Table 1. The low- and middle-income countries (LMICs) context emerged from the SLR.

**Table 1.** Classification and coding framework.

| Classification | Description | Codes |
| --- | --- | --- |
| Context | Global | 1A |
| | Sub-Saharan Africa | 1B |
| | South Africa | 1C |
| | Low- and middle-income countries | 1D |
| Focus | Digital platform | 2A |
| | Business model innovation | 2B |
| | Digital literacy or skills | 2C |
| | Digital infrastructure | 2D |
| | 4IR | 2E |
| | Affordability | 2F |
| Method | Qualitative | 3A |
| | Quantitative | 3B |
| | Theoretical | 3C |
| | Empirical | 3D |
| | Case studies/interviews | 3E |
| | Survey | 3F |
| | Design | 3G |
| Sector | Agriculture | 4A |
| | Not applicable | 4B |
| Origin (continents) | America | 5A |
| Africa | Europe | 5B |
| | Asia | 5C |
| | Africa | 5D |
| | Oceania | 5E |

## 3. Results and Analysis of the Systematic Literature Review

The 42 selected research articles were coded according to the coding and classification framework in Table 1 with the resultant coding results shown in Supplementary Material, Table S1. Summary of 42 selected articles main objectives is provided in Supplementary Material Table S2 (Refer to the Supplementary Material Table S2). A detailed analysis of the findings in accordance with the coding and classification framework (context, focus, method, sector, and origins) is provided in Section 3.1, Section 3.2, Section 3.3, Section 3.4, Section 3.5 to help answer RQ1. Section 3.6 provides a detailed analysis of the dimensions of interest as they emerged from the SLR to help answer RQ2, RQ3, RQ4, RQ5 and RQ6.

### 3.1. Context

The context analysis was explored based on the research that has been conducted at a global, sub-Saharan African, and South African level. There were 25 articles identified from the global level, 12 articles were from sub-Saharan African countries, while 4 articles were from South Africa, and 1 article was from an LMIC (Table 2). The 25 articles at a global level originated from Europe; Central, South, and North America; East, South, Southeast, and Western Asia; and Oceania (Italy, India, Taiwan, Sri Lanka, Indonesia, Portugal, Vietnam, Colombia, Pakistan, Cambodia, Philippines, Thailand, Greece, Spain, Serbia, United States of America, China, Myanmar, Iran, Lithuania, Brazil, Netherlands, Germany, and Austria). The studies that came from sub-Saharan countries were from

Kenya, Ethiopia, Tunisia, South Africa, Malawi, Mali, Nigeria, Tanzania, Uganda, Ghana, Mozambique, and Tanzania.

**Table 2.** Context.

| Context | No. of Articles | Research Context Code |
|---|---|---|
| Global level | 25 | 1A |
| Sub-Saharan Africa level | 12 | 1B |
| South Africa level | 4 | 1C |
| Low- and middle-income countries level | 1 | 1D |

The studies are geographically limited in context, and the findings are limited in terms of universal validity [31,32]. Alemayehu and Van Vuuren (2017) [33] identified that small businesses in sub-Saharan Africa are facing systemic challenges that are exclusive to them and not shared by enterprises in other countries. These challenges include limitations in accessing financial resources, markets and marketing skills, and the business environment. It is, therefore, important to conduct theoretical and empirical research that considers local contexts using geographical location, culture, and profiles of enterprises as categorized by [31].

The overall findings suggest that the intensity of research in relation to digital agriculture ecosystems for smallholder farmers in the context of sub-Saharan Africa is very low when compared with research performed at a global level. This presents a significant challenge for smallholder farmers in such contexts to leverage digital agriculture ecosystems for sustainability, competitive advantage, and market growth. Even though there is one study focusing on LMICs, it did not explicitly mention where the study was conducted. These findings suggest that studies on digital agriculture ecosystems are, in general, limited in sub-Saharan Africa and even more so in South Africa. Further studies that explicitly consider sub-Saharan Africa and South Africa are needed to investigate digital agriculture ecosystems in the context of smallholder farmers. It is, therefore, recommended that research be conducted on digital agriculture ecosystems in relation to smallholder farmers that explicitly considers sub-Saharan Africa in general and South Africa in particular.

*3.2. Continents of Origin*

The majority of the research originated from Africa (20) and Asia (13); 8 studies originated in Europe, while 6 originated from America, and only 2 originated from Oceania. While the majority of research originated in Africa, only four studies were conducted in South Africa. To help with making more research available on digital agriculture ecosystems for smallholder farmers in sub-Saharan Africa, it is recommended that research be carried out taking into account the contextual challenges of smallholder farmers.

*3.3. Focus on Digital Ecosystems for Smallholder Farmers*

Out of the 42 articles, including the combinations, 2 articles researched digital platforms (2A), 4 articles researched digital infrastructure (2D), and 15 articles researched 4IR (2E). There was only one article that researched business model innovation (2B). There were no articles that researched digital literacy or skills (2C) nor affordability (2F). These findings suggest that the majority of research focuses on three components of a digital agriculture ecosystem (digital platforms, digital infrastructure, and 4IR technologies), with very limited research directed at digital literacy or skills, affordability, and business model innovation. This means that there is no research yet that focuses on all the components of a digital agriculture ecosystem. These findings indicate that research on digital agriculture ecosystems for smallholder farmers is still a nascent and emerging research area with a very limited focus on the specific components of such a system. This is a research gap that presents significant opportunities that need to be further explored, as the successful

adoption of digital agriculture by smallholder farmers is contingent on the existence of a fully functional digital agriculture ecosystem within the smallholder farmer context and environment.

### 3.4. Research Methods

There was diversity in the use and application of research methods in the 42 research articles. Authors used single, mixed, qualitative, and quantitative research methods, which included qualitative (3A), quantitative (3B), theoretical (3C), empirical (3D), case study (3E), survey (3F), and design (3F) approaches for data collection (Table 3). Only 5 of the 42 articles utilized empirical methods. The findings indicate the need for research to use more empirical methods that can contextualize the digital agriculture ecosystems research to smallholder farmers' specific contexts. Specifically, empirical research utilizes qualitative research methods as they bring out context-specific challenges, constraints, issues, and preconditions.

**Table 3.** Research methods analysis.

| Research Methods | Qualitative 3A | Quantitative 3B | Theoretical 3C | Empirical 3D | Case Studies 3E | Survey 3F | Design 3G | No. of Articles |
|---|---|---|---|---|---|---|---|---|
| Single research methods | 0 | 0 | 7 | 0 | 0 | 0 | 4 | 11 |
| Mixed research methods | 5 | 5 | 1 | 1 | 2 | 2 | 0 | 16 |
| Qualitative | 13 | 0 | 2 | 1 | 11 | 4 | 1 | 32 |
| Quantitative | 0 | 12 | 0 | 3 | 8 | 8 | 2 | 33 |

### 3.5. Current Research Performed on Digital Ecosystems for Smallholder Farmers

The articles were analyzed to determine the current research conducted on digital agriculture ecosystems in relation to smallholder farmers to further answer RQ1. Out of the 42 identified research articles, there were no studies that focused on all the elements of a digital agriculture ecosystem. The research studies focused on 4IR (2E) technologies such as big data, AI, blockchain, the IoT, sensors, and cloud computing.

The studies also focused on digital platforms (2A), directing attention toward, for example, farm management platforms such as soil and plant monitoring, yield monitoring, predictive data analysis systems, breeding management, and real-time variable-rate fertilizer application systems. Digital platforms research also focused on platforms such as myAcker (for remote online gardeners to virtually manage agricultural plots in a game-like setting); the web platform SIRIA (http://siria.fenalce.org/ accessed on 4 June 2022); Agri-Cloud (South Africa); MAP (Modern Agriculture Platform Chongzhou); AMIS (Malawi Agriculture Market Information System); decision support systems; IoT farming platforms; integrated pest management; and online dashboards; as well as on mobile applications, including mobile money for agricultural use: namely, mKesh, Green Way Agri-Livestock, Site Pyo—Cultivation, Golden Paddy, Htwet Toe, and Plant Protection. Lastly, the research also focused on several digital platforms for agriculture use, such as Kilimo Salama, M-PESA and M-Shwari (Kenya), AgiLife (Kenya and Uganda), Farm Force, E-Wallet, M-PESA, the web-based platform Esoko, the Kenya-based platform Sokopepe, Twiga Food in Kenya, the Mergata mobile-based platform, Mastercard 2KUZE, Ujuzi Kilimo, EthioSIS, and Phytclean.

Research on digital infrastructure (2D) focused on information and communication technology (ICT), both wired and wireless, mobile cellular and mobile broadband connection, as well as the Internet. Only one article focused on business model innovation (2B), and there was no research performed on digital literacy or skills (2C) or affordability (2F). This presents an opportunity for further research to be conducted on a complete digital agriculture ecosystem as it applies to smallholder farmers, specifically research that will consider digital literacy or skills, affordability, and business model innovation elements. Smallholder farmers, just like small, micro, and medium enterprises (SMMEs),

are financially constrained and lack the human resources, required skills, and business model innovation for the adoption of digital solutions.

### 3.6. Dimensions Emerging from the Systematic Literature Review

The following sections answer research questions RQ2, RQ3, RQ4, RQ5, and RQ6. Based on these questions the following dimensions emerged from the SLR: (1) challenges, (2) level of access to and uptake, (3) use, and (4) benefits. The dimensions were analyzed in accordance with the codification results in Supplementary Material, Table S1.

3.6.1. What Are the Challenges of Digital Agriculture Ecosystems in Relation to Smallholder Farmers?

RQ2 sought to understand the current challenges of digital agriculture ecosystems in relation to smallholder farmers. Out of the 42 articles selected for the SLR, 32 explicitly mentioned and discussed the challenges of existing digital solutions for smallholder farmers, while 10 did not mention or discuss any challenges.

The identified challenges were categorized according to the elements of a digital ecosystem for agriculture; that is, digital platform (2A), business model innovation (2B), digital literacy or skills (2C), digital infrastructure (2D), 4IR (2E), and affordability (2F) (Figure 4). There were no challenges that were mentioned or discussed regarding digital platforms (2A). More than 50% of the challenges concerned digital infrastructure (2D), followed by the challenges related to affordability (2F) (more than 75%). The majority of challenges (more than 90%) regarding a digital ecosystem for agriculture concerned digital literacy or skills (2C).

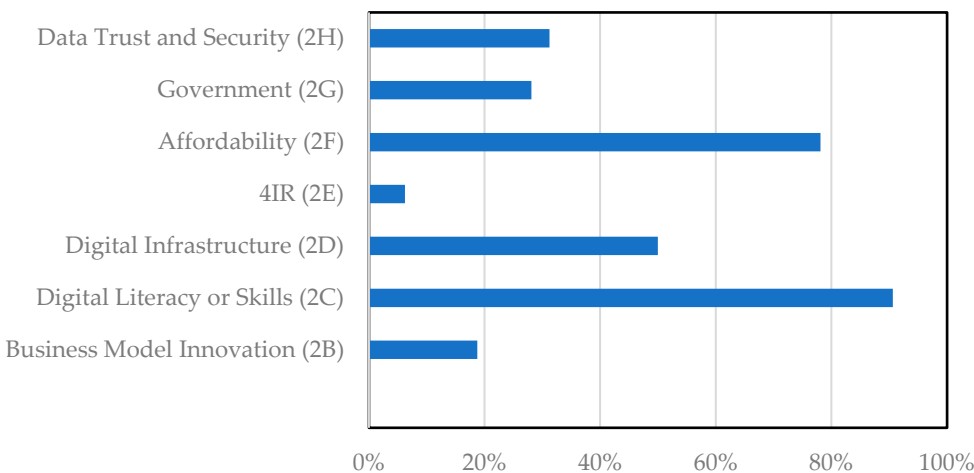

**Figure 4.** Challenges under each element of a digital ecosystem for agriculture.

As shown in Table 4, challenges under digital literacy or skills (2C) include inadequate information about digital solutions, digital agriculture being a new concept to smallholder farmers, a lack of perceived value, timely and relevant information, language barriers, first generation in farming (the Old), and a lack of knowledge about digital solutions. The lack of perceived value of digital solutions may be the reason why the challenges related to business model innovation (2B) are very low. The challenges of digital infrastructure (2D) include land tenure issues, telecommunication and electrical infrastructure, and remote location. Affordability (2F) challenges emanate from data rates and/or communication costs, the costs of using technology, maintenance, replacing labor, and a lack of financial and human resources. While challenges related to 4IR (2E), government (2G), and data trust and security (2H) were very low, there were important factors of significant impact that were identified. Cybersecurity was identified as a key challenge regarding 4IR (2E) technologies, while legal infrastructure, political and social ramifications, and standardization and interoperability challenges were key factors for government (2G). Trustworthiness emerged as a key factor

for data and trust security (2H). The findings reflected in Table 4 indicate that digital ecosystems for agriculture are still too underdeveloped for smallholder farmers to be able to consider digital agriculture as a viable solution to their needs and requirements. Research focusing on the identified challenges is needed to support the development of a viable digital agriculture ecosystem for smallholder farmers.

**Table 4.** Factors influencing challenges of digital solutions for smallholder farmers.

| Elements of Digital Agriculture Ecosystem | Factors of Influence (Challenges) |
|---|---|
| Business model innovation (2B) | • Lack of perceived value of digital solutions |
| Digital literacy or skills (2C) | • Inadequate information about digital solutions<br>• Digital agriculture is a new concept for smallholder farmers.<br>• Lack of perceived value<br>• Timely and relevant information<br>• Language barriers<br>• First generation in farming (the Old)<br>• Lack of knowledge about digital solutions |
| Digital infrastructure (2D) | • Land tenure issues<br>• Telecommunication and electrical infrastructure<br>• Remote location |
| 4IR (2E) | • Cybersecurity |
| Affordability (2F) | • Data rates and/or communication costs<br>• Cost of using technology<br>• Maintenance and replacing labor<br>• Lack of financial and human resources |
| Government (2G) | • Legal infrastructure<br>• Political and social ramifications<br>• Standardization and interoperability |
| Data trust and security (2H) | • Trustworthiness |

### 3.6.2. How Are Smallholder Farmers Using Digital Solutions in Their Businesses?

RQ3 sought to establish how smallholder farmers are using digital solutions in their businesses. Of the articles, 24 mentioned and explicitly discussed the use of digital solutions by smallholder farmers, while 18 did not mention or discuss such use.

Smallholder farmers are using digital solutions mainly for accessing agricultural information (46%) and monitoring and tracing (46%), followed by selling and/or marketing of products (29%), detection (25%), and planting and production (21%) (Figure 5). Although to a limited extent, smallholder farmers are also using digital solutions for fertilization, communication, buying input supplies, harvesting, irrigation, banking/saving money, controlling farming equipment, collecting agricultural data, refrigeration, mapping, prediction, and for client creditworthiness assessment, as shown in Figure 5.

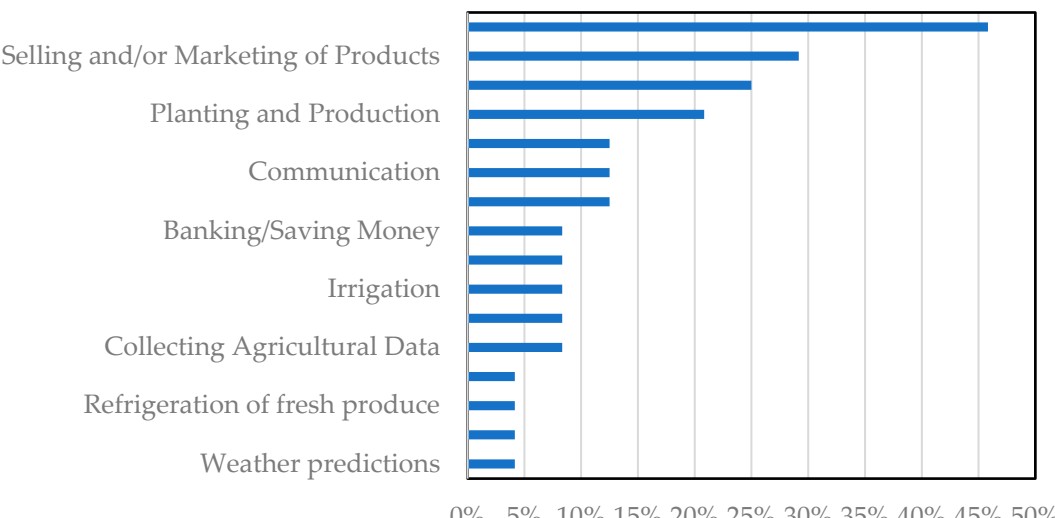

**Figure 5.** Use of digital solutions by smallholder farmers.

Smallholder farmers are accessing agricultural information about livestock and livestock management techniques, agronomy, weather forecasts, forecasts on maize planting, recommendations by fellow farmers for their specific farm locations, farming practices, daily crop market trends and prices, and market information. Farmers are also accessing agriculture information on possible occurrences of plant or crop diseases; pests; sudden floods; unseasonal rain, wind, droughts, and other warning notifications; as well as the best time to harvest crops.

Smallholder farmers use digital solutions for monitoring and tracing to police stock theft and survey different areas of their farms. Furthermore, digital solutions are used to keep a close eye on the location, health, and general well-being of their cattle; find lost or stolen cattle; scare away predator animals; for cattle herding; to monitor ill, injured, or distressed cattle; and for production efficiency. Lastly, they are used in resolving crop problems and/or selling and providing information on the state of the soil and surrounding environment in terms of pH, moisture, texture, color, air temperature, and light.

Smallholder farmers use the accessed agricultural information to gain knowledge on daily crop market trends and prices and on market information for required interventions such as planting more crops for which they have price information, resolving crop problems, and selling. Refrigeration is used to preserve the shelf life and quality of fruits and vegetables along with perishable agro-processing products until they reach supermarket shelves; to monitor and detect the freshness of produce; to delay the ripening of fruit; to automate processes in the supply chain; and to acquire, manage, store, and share information and knowledge.

The smallholder farmers are also using digital solutions to detect possible occurrences of plant or crop diseases, pests, sudden floods; unseasonal rain, wind, droughts, and other warning notifications; the best time to harvest crops; and to detect information on the state of the soil and surrounding environment in terms of pH, moisture, texture, color, air temperature, and light.

These findings suggest that, although digital agriculture is still a nascent concept to smallholder farmers, a small segment of these farmers is already using digital solutions for various applications, such as accessing information related to agriculture and the selling and marketing of agricultural products. Research focusing on further investigating the applicability, relevancy, and value proposition of these applications in various smallholder farmer settings and contexts will provide opportunities for the generalization and validity of these applications.

### 3.6.3. What Are the Factors That Influence the Uptake of Digital Solutions by Smallholder Farmers?

RQ4 sought to establish the factors that influence the uptake of digital solutions by smallholder farmers. Thirty-one articles mentioned and explicitly discussed the factors that influence the uptake of digital solutions by smallholder farmers, while eleven articles did not do so.

The factors that were identified as influencing the uptake of digital solutions were categorized according to the elements of a digital agriculture ecosystem: namely, digital platform (2A), business model innovation (2B), digital literacy or skills (2C), digital infrastructure (2D), 4IR (2E), and affordability (2F) (Figure 6). The study did not find any enabling factors linked to digital platform (2A) or 4IR (2E) elements (Figure 6).

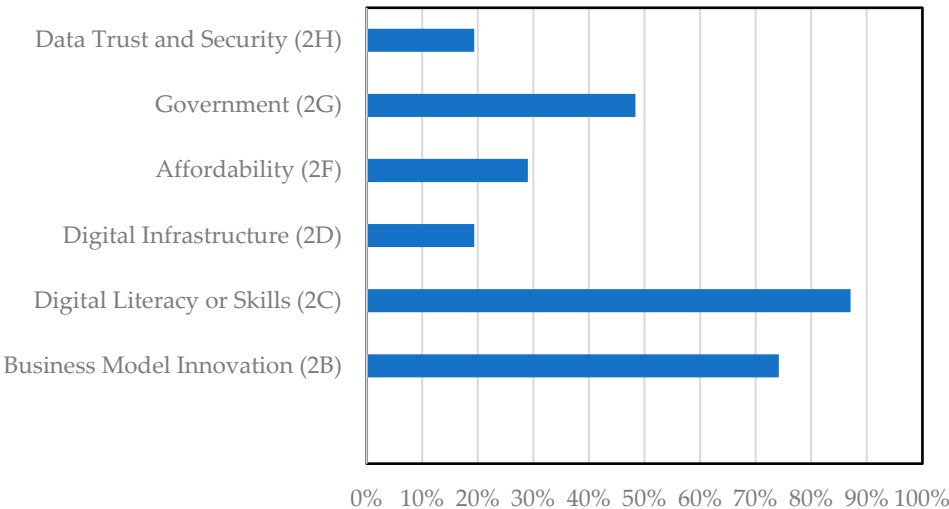

**Figure 6.** Factors enabling digital solutions uptake by smallholder farmers.

The primary factor influencing the uptake of digital solutions by smallholder farmers is digital literacy or skills (2C) (87%). Digital literacy or skills can be improved through timely and up-to-date data and knowledge, creating awareness, changing perceptions and behaviors, specific training for older farmers, understanding from farmers how digital solutions should be introduced to them, understanding the knowledge and attitude status of farmers, the inclusion of youth in agricultural education and training, smart agriculture knowledge, and business knowledge and capabilities (Table 5).

Digital literacy or skills (2C) is followed by business model innovation (2B) (74%). Business models can be innovated by offering off-the-shelf or do-it-yourself (DIY) digital solutions and through enhanced usability/ease of use; reproducibility of offered digital solutions; government and private sector partnerships (public–private partnerships); partnerships with community-based farmer associations; gender-responsive digital solutions; localized or contextualized digital solutions; and by ensuring the usefulness and utility of the offered digital solutions, as shown in Table 5.

Government (2G) (48%) also emerges as a key enabler for the uptake of digital solutions by smallholder farmers; specifically, the need for the development and implementation of policies that are geared toward supporting smallholder business environments and contexts (Table 5).

**Table 5.** Factors influencing the uptake of digital solutions.

| Digital Agriculture Ecosystem Elements | Factors of Influence |
|---|---|
| Business model innovation (2B) | <ul><li>Offering off-the-shelf or DIY digital solutions</li><li>Enhanced usability/ease of use and reproducibility of offered digital solutions</li><li>Government and private sector partnerships (public–private partnerships)</li><li>Partnerships with community-based farmer associations</li><li>Gender-responsive digital solutions</li><li>Localized or contextualized digital solutions</li><li>Usefulness and utility of offered digital solutions</li></ul> |
| Digital literacy or skills (2C) | <ul><li>Timely and up-to-date data and knowledge</li><li>Creating awareness</li><li>Changing perceptions and behaviors</li><li>Older farmers' specific training</li><li>Understanding from farmers how digital solutions should be introduced to them</li><li>Understanding the knowledge and attitude status of farmers</li><li>Inclusion of youth in agriculture education and training</li><li>Smart agriculture knowledge</li><li>Business knowledge and capabilities</li></ul> |
| Digital infrastructure (2D) | <ul><li>Connectivity and mobile broadband access</li><li>Availability of a suitable spectrum</li></ul> |
| Affordability (2F) | <ul><li>Credit access</li><li>Access to mobile savings accounts</li><li>Affordable digital solutions</li></ul> |
| Government (2G) | <ul><li>Development and implementation of policies that are geared toward supporting the smallholder business environment and context</li></ul> |
| Data trust and security (2H) | <ul><li>Establishing trust among the smallholder farming community</li></ul> |

Affordability (2F) (29%) is influenced by factors such as credit access, access to mobile savings accounts, and affordable digital solutions, while digital infrastructure (2D) (19%) is influenced by connectivity and mobile broadband access and the availability of a suitable spectrum for the uptake of digital solutions by smallholder farmers, as shown in Table 5.

Lastly, the findings reveal that for data trust and security (2H) (19%), establishing trust among the farming community is one of the most important factors influencing the uptake of digital solutions by smallholder farmers (Table 5).

3.6.4. What Are the Benefits of Digital Solutions for Smallholder Farmers?

RQ5 sought to establish the benefits of digital solutions for smallholder farmers. While 28 articles explicitly mentioned and discussed the benefits of digital solutions for smallholder farmers, the remaining 14 articles did not do so.

Of the 28 articles that identified benefits of using digital solutions for smallholder farmers, 10 mentioned getting agriculture information such as real-time prices; timely alerts on weather, market trends, pests, and diseases; damage identification; and advice on pesticide and fertilizer use as shown in Supplementary Material, Table S3. Nine articles mentioned gaining better control and precision as a benefit of using digital solutions for smallholder farmers to improve water conservation and to reduce soil erosion, environmen-

tal impact, and agricultural inputs. Other main identified benefits of using digital solutions are productivity (8 articles), better earning and higher yield (7 articles), improving product quality (7 articles), lowering costs or reducing transactional costs (6 articles), access to markets (5 articles), resilient farm production (4 articles), suppliers and value chain relations (4 articles), and quality standards and compliance (4 articles) (Refer to Supplementary Material, Table S3).

The systematic review also revealed efficiency, sustainability, transparency, a positive impact on agricultural income, real-time data collection, improving farming decisions and predictions, cash management, and financial identity as benefits of using digital solutions for smallholder farmers. Other interesting benefits of using digital solutions for smallholder farmers are a slowdown in migrating to cities [34], obtaining farming-related advice in one's own language [35], new knowledge generation [36–39], and women's empowerment [37].

The research findings point to the importance of digital solutions for smallholder farmers. This suggests that the time is "ripe" for research to be conducted on digital agriculture ecosystems for smallholder farmers to ensure that these farmers are not left out of the benefits that come with using digital solutions. The findings also point to the need for research to explicitly focus on identifying the benefits of using digital solutions for smallholder farmers. Future research should investigate whether taking the identified benefits of digital solutions to smallholder farmers and incorporating them into a digital agriculture ecosystem framework could translate into a potential relative advantage for smallholder farmers. Lastly, the benefits identified here need to be communicated to smallholder farmers to accelerate the uptake of digital solutions.

### 3.6.5. What Is the Level of Access to and Uptake of Digital Solutions by Smallholder Farmers?

RQ6 sought to establish the level of access and uptake of digital solutions by smallholder farmers. Eighteen articles mentioned and discussed the level of access to and uptake of digital solutions by smallholder farmers, while twenty-four articles did not do so.

Of the 18 articles, 3 indicate that the level of access to and uptake of digital solutions by smallholder farmers is still very limited [40–42]. Technologies such as wireless sensor networks, which are critical for digital agriculture, are indicated as not being widely adopted globally, especially by small-scale farmers [34]. Although the level of access and uptake of digital solutions by smallholder farmers is still very limited, smallholder farmers have been gradually increasing their level of access to and uptake of farm management software; data analytics and forecasting applications; smart irrigation systems; soil, plant, and crop monitoring systems; precision livestock systems; smart agriculture systems; drones and robotic farm applications; smart greenhouse management systems; cloud computing applications; environmental control systems; and mobile phones.

The majority (63%) of the 18 articles focused on mobile phones, while the other 11 articles were limited to a few digital innovations, as mentioned. These findings indicate that there is limited research focused on determining the level of access to and uptake of digital solutions by smallholder farmers. The existing research concerns only a few digital innovations or technologies in relation to digital solutions for smallholder farmers. There is thus an opportunity for research to address this gap, which is key to establishing the status quo and tracking progress in relation to the adoption of digital agriculture by smallholder farmers. A clear understanding of the status quo will assist all involved stakeholders (such as academia, industry, practitioners, and government) to know whether current interventions are yielding any positive outcomes or whether other approaches are needed.

## 4. Recommendation

This section provides recommendations for future research agendas that came out as a result of the research findings that resulted from this systematic literature review. Based on the findings and recommendations under context, the following research question is posed:

- How can a digital agriculture ecosystem be developed for smallholder farmers in the agriculture sector of sub-Saharan Africa?

Based on the findings and recommendations on the origin of the studies, the following research questions are asked:

- How can a digital agriculture ecosystem that considers contextual challenges be developed for sub-Saharan smallholder farmers?
- What is the definition of smallholder farmers in the context of the sub-Saharan agriculture sector?

To help with addressing the identified research gaps under focus on the digital agriculture ecosystem for smallholder farmers, the authors propose the following research questions:

- What is the level of digital literacy or skills of smallholder farmers?
- Do smallholder farmers have access to digital infrastructure?
- What is the impact of the digital infrastructure on digital solutions uptake by smallholder farmers?

The following future research question can be asked to assist with research that focuses on all elements of a digital agriculture ecosystem (digital platforms, digital infrastructure, business model innovation, digital literacy or skills, affordability, and 4IR technologies) to add to the body of knowledge:

- How can a digital agriculture ecosystem for smallholder farmers be developed that focuses on all the elements of the ecosystem and considers contextual challenges facing these farmers?

The following questions could be asked to assist, under the ambit of government, with alleviating the specific challenges identified:

- How can government develop a supportive legal infrastructure for smallholder farmers to enable digital transformation?
- What are the political and social ramifications impeding smallholder farmers from taking up and using digital solutions?
- How can government assist with the standardization and interoperability of digital solutions for smallholder farmers?

The questions to be asked to help with challenges identified under digital skills (2C) and affordability (2F) are:

- How can a digital solution for smallholder farmers' development incorporate the local languages of smallholder farmers?
- How can a digital solution for smallholder farmers be developed in a way that considers first-generation farmers?
- What is needed to educate smallholder farmers on developing digital solutions?
- How can the value of digital solutions for smallholder farmers be communicated to them?
- What can be done to bring the cost of communication and data rates down for smallholder farmers?

The questions to be asked to assist with eliminating the challenges identified under business model innovation (2B) and digital infrastructure (2D) are as follows:

- How can digital solutions for smallholder farmers be developed in a way that increases the participation of women in smallholder farming?
- How should digital solutions be communicated in a way that creates awareness for smallholder farmers?
- How do digital solutions for smallholder farmers consider agriculture extension services?
- What is the impact of land tenure issues on the uptake of and access to digital solutions by smallholder farmers?
- How should digital solutions for smallholder farmers be developed in a way that considers land tenure issues?

- What is the effect of telecommunication and electrical infrastructure on digital solutions for smallholder farmers?
- How should digital solutions for smallholder farmers be developed in a way that considers the challenges posed by telecommunication and electrical infrastructure?
- How can digital solutions for smallholder farmers be developed in a way that takes into account the remote locations of farmers?

Based on the key findings on how smallholder farmers are using digital solutions in their businesses, the following research question is posed:

- What is the applicability, relevancy, and value proposition of digital solutions to various smallholder farmer settings and contexts?

Recommendations for future research agendas that resulted from RQ1 to RQ6 are presented below.

Research questions to help with identifying factors of influence under digital literacy or skills are:

- How do smallholder farmers want digital solutions to be introduced to them?
- How should awareness of digital solutions for smallholder farmers be created?
- How can timely, up-to-date data and knowledge about digital solutions for smallholder farmers be achieved through a digital agriculture ecosystem?
- How can the perceptions and behaviors of smallholder farmers toward digital solutions be changed?
- How can training on digital solutions be developed and provided for older smallholder farmers?
- What is the level of knowledge and attitudes of smallholder farmers toward digital solutions?
- How can the inclusion of youth in agriculture education and training on digital solutions for smallholder farmers be achieved?
- What is the level of knowledge of smallholder farmers on digital innovation within digital agriculture solutions?
- Do smallholder farmers have the business knowledge and capabilities required for them to uptake digital solutions?

Research questions to help with identifying factors of influence in relation to business model innovation are:

- How can digital solutions be contextualized for the smallholder farmers' contexts?
- How can business model innovation assist with gender-responsive digital solutions for smallholder farmers?
- How can developed digital solution offerings be made available off-the-shelf or through DIY for smallholder farmers?
- How can the design of digital solutions for smallholder farmers be developed to ensure enhanced usability/ease of use and reproducibility?
- What role can government and private sector partnerships (public–private partnerships) and partnerships with community-based farmers' associations play in the development of digital solutions for smallholder farmers?
- What is the usefulness and utility of the offered digital solutions for smallholder farmers?

A research question to help with identifying factors of influence affecting government is:

- How can government develop and implement policies that are geared toward supporting smallholder business environments and contexts?

Research questions to help with identifying factors of influence affecting affordability are:

- How can digital solutions for smallholder farmers be developed in such a way that they are affordable for smallholder farmers?
- How can the development of digital solutions ensure access to mobile savings accounts and credit?

Research questions to help with identifying factors of influence related to digital infrastructure are:

- What solutions can be put in place to ensure connectivity and mobile broadband access to enable the uptake and use of digital solutions by smallholder farmers?
- How is the availability of spectrum affecting the access to and uptake of digital solutions by smallholder farmers?

A research question to help with identifying factors of influence affecting data trust and security is:

- How can trust in digital solutions be established among the smallholder farming community?

Questions to assist with unearthing the benefits of digital solutions for smallholder farmers could be:

- How should the benefits of digital solutions be communicated to smallholder farmers to facilitate the uptake of such solutions?
- How does one develop a digital agriculture ecosystem so that it delivers the identified benefits of using digital solutions for smallholder farmers?
- How can the age of older smallholder farmers be taken into account when communicating the benefits of digital solutions to such farmers?
- How should the benefits of digital solutions be communicated to young people when research has revealed that they are not interested in farming?
- How can the benefits of digital solutions be communicated in a way that attracts young people to farming?

The recommended future research questions to assist in unearthing the level of access to and uptake of digital solutions by smallholder farmers could be:

- Which digital agriculture platforms do smallholder farmers have access to and which are they taking up?
- What is the level of access to and uptake of agricultural business model innovations by smallholder farmers?
- What is the level of access to digital infrastructure by smallholder farmers?
- What is the level of access to and uptake of 4IR technologies by smallholder farmers?
- How is affordability affecting the access to and uptake of digital solutions by smallholder farmers?

## 5. Conclusions

Smallholder farmers are the primary source of income, employment, poverty alleviation, and livelihoods in remote rural areas. Yet, smallholder farmers are highly constrained by the specific contextual challenges of climate change, productivity, cost of production, credit access, and financial resources constraints, all of which impact their sustenance, sustainability, and growth. Digital agriculture is seen as a viable solution to addressing smallholder farmers' contextual challenges. However, many smallholder farmers are beyond the reach of these digital solutions due to underdeveloped or nonexistent digital ecosystems for smallholder farmers.

The key findings of this systematic review suggest that research on digital agriculture ecosystems is very limited in sub-Saharan Africa. Most research focuses only on digital platforms, digital infrastructure, and 4IR technology components of the digital agriculture ecosystems, with very limited research directed at digital literacy or skills, affordability, and business model innovation.

The findings also reveal that digital ecosystems for agriculture are still too underdeveloped for smallholder farmers to be able to consider digital agriculture as a viable solution to their needs and requirements. As a result, only a few smallholder farmers are currently using digital solutions in their businesses to access agricultural information and to sell and market their agricultural produce.

The findings reveal a research gap that calls for further studies, as the successful adoption of digital solutions by smallholder farmers is contingent on the existence of fully functional digital agriculture ecosystems within the smallholder farmers' contexts and environments. As a result, this article has drawn out an agenda for future research by suggesting new questions in the six thematic areas (digital platforms, digital infrastructure, business model innovation, digital literacy or skills, affordability, 4IR technologies, and data trust and security) to close the research gap.

Finally, this study utilized only four multidisciplinary databases as the data sources for the SLR, and this may have affected the number of relevant papers found. More multidisciplinary databases should be considered for future systematic reviews on the digital ecosystems for smallholder farmers.

**Supplementary Materials:** The following supporting information can be downloaded at: https://www.mdpi.com/article/10.3390/su151612530/s1, Table S1: Results of Articles Codification; Table S2: Brief Summary of Articles' Main Objectives and Results. Table S3: Benefits of using digital solutions for smallholder farmers.

**Author Contributions:** Conceptualization, N.G.; methodology, N.G.; validation, L.G., H.T. and N.G.; formal analysis, N.G.; investigation, N.G.; resources, H.T.; data curation, L.G. and H.T.; writing—original draft preparation, N.G.; review and editing N.G., L.G. and H.T.; visualization, N.G.; supervision, H.T. All authors have read and agreed to the published version of the manuscript.

**Funding:** This research received no external funding.

**Institutional Review Board Statement:** Not applicable

**Informed Consent Statement:** Not applicable

**Data Availability Statement:** Not applicable

**Conflicts of Interest:** The authors declare no conflict of interest.

## Appendix A. Specific Terms and Search Strings

The specific terms for the [**Unit of Analysis**] were:

- Smallholder Farmers OR
- Small-Scale Farmers OR
- Subsistence Farmers OR
- Resource Poor Farmers OR
- Low-Input Farmers OR
- Low-Income Farmers OR
- Small Micro and Medium Enterprises Farmers OR
- SMMEs Farmers OR
- Small Business Farmers OR
- Small and Medium Sized Enterprise Farmers OR
- SMEs Farmers OR
- Small and Medium Business Farmers OR
- SMBs Farmers

The specific terms for the [**Technology Artifact**] were:

- Digital Ecosystem OR
- Artificial Intelligence OR
- Cloud Computing OR
- Internet of Things OR
- Smart Sensors OR
- Remote Sensing OR
- Big Data OR
- Mobile

The specific terms for the [**Phenomenon of Interest**] were allowed to emerge from the systematic literature search.

The [**Technology Artifacts**]:

1. Digital Ecosystem OR Artificial Intelligence OR Cloud Computing OR Internet of Things OR Smart Sensors OR Remote Sensing OR Big Data OR Mobile

The [**Unit of Analysis**] AND [**Technology Artifact**]:

2. Smallholder Farmers AND Digital Ecosystem OR Artificial Intelligence OR Cloud Computing OR Internet of Things OR Smart Sensors OR Remote Sensing OR Big Data OR Mobile

3. Small-Scale Farmers AND Digital Ecosystem OR Artificial Intelligence OR Cloud Computing OR Internet of Things OR Smart Sensors OR Remote Sensing OR Big Data OR Mobile

4. Subsistence Farmers AND Digital Ecosystem OR Artificial Intelligence OR Cloud Computing OR Internet of Things OR Smart Sensors OR Remote Sensing OR Big Data OR Mobile

5. Resource Poor Farmers AND Digital Ecosystem OR Artificial Intelligence OR Cloud Computing OR Internet of Things OR Smart Sensors OR Remote Sensing OR Big Data OR Mobile

6. Low-Input Farmers AND Digital Ecosystem OR Artificial Intelligence OR Cloud Computing OR Internet of Things OR Smart Sensors OR Remote Sensing OR Big Data OR Mobile

7. Low-Income Farmers AND Digital Ecosystem OR Artificial Intelligence OR Cloud Computing OR Internet of Things OR Smart Sensors OR Remote Sensing OR Big Data OR Mobile

8. Small Micro and Medium Enterprise Farmers AND Digital Ecosystem OR Artificial Intelligence OR Cloud Computing OR Internet of Things OR Smart Sensors OR Remote Sensing OR Big Data OR Mobile

9. SMMEs Farmers AND Digital Ecosystem OR Artificial Intelligence OR Cloud Computing OR Internet of Things OR Smart Sensors OR Remote Sensing OR Big Data OR Mobile

10. Small Business Farmers AND Digital Ecosystem OR Artificial Intelligence OR Cloud Computing OR Internet of Things OR Smart Sensors OR Remote Sensing OR Big Data OR Mobile

11. Small and Medium Sized Enterprise Farmers AND Digital Ecosystem OR Artificial Intelligence OR Cloud Computing OR Internet of Things OR Smart Sensors OR Remote Sensing OR Big Data OR Mobile

12. SMEs Farmers AND Digital Ecosystem OR Artificial Intelligence OR Cloud Computing OR Internet of Things OR Smart Sensors OR Remote Sensing OR Big Data OR Mobile

13. Small and Medium Business Farmers AND Digital Ecosystem OR Artificial Intelligence OR Cloud Computing OR Internet of Things OR Smart Sensors OR Remote Sensing OR Big Data OR Mobile

14. SMBs Farmers AND Digital Ecosystem OR Artificial Intelligence OR Cloud Computing OR Internet of Things OR Smart Sensors OR Remote Sensing OR Big Data OR Mobile.

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
