# Peer review of "Towards Sustainable Digital Agriculture for Smallholder Farmers: A Systematic Literature Review"

_sustainability, doi:10.3390/su151612530_

Round 1
Reviewer 1 Report
The review article entitled "Towards sustainable digital agriculture for smallholder farmers" is well analyzied and written by the authors. The methodlogy and protocols defined and used for the analysis is fine and comprehensively described and well executed. The articles reviewed are from around the globe and balanced.
Results are presented in tangible fashion and well written.
Quality of english language is fine. minor spell check may be considered.
Author Response
Please see attached cover letter explaining how reviewer's comments were addressed.

Reviewer 2 Report
Guidance to small holding farmers is important for food security in developing countries. Present article describes an insight of digital solutions for small farmers. Information is gathered from published articles is constructed well, however, needs english improvement.
-Abstract is written well.
-Introduction section provides much of information about digital agriculture and background of report. However, there are too long sentences those must be rewritten.
-Mathodology, results, discussion written comprehensively well.
There are long scentences, which needs to be rewritten.
Author Response
Please find attached cover letter explaining how Reviewer's comments were addressed.

Reviewer 3 Report
The authors presented an interesting review work on sustainable digital agriculture for smallholder farmers. Although, the topic is very valuable and within the scope of Sustainability, there are some major concerns on the presentation, especially the results and discussion section. The listed suggestions are for the authors to further improve their work prior to its recommendation for publication.
Line 198: … Sub-Saharan Africa in general and South Africa in particular.
Line 205. Studies, respectively.
How many research questions does the current study specifically seek to address? I understand that the authors also proposed some research questions. But more attention should be focused on the main research questions the current study targeted at addressing.
Also, some of the items presented under the results and analysis of the systematic literature review are not appropriate under this section. For instance, section 3.4 research methods.
A section on recommendations for future direction should also be provided.
The English language quality of the presentation is fine. Only moderate revision is required.
Author Response

(The authors gave the same response as above.)

Reviewer 4 Report
My main comments and suggestions:
1. The title should be changed as it does not follow from the text of the manuscript and does not indicate that it is a review. In addition, it should be specified what type of review: critical, systematic, overview.
2. I suggest clearly specifying the goals of the work. There are many hypotheses and questions in the manuscript that make it difficult to read.
3. Delete paragraph line 93-95.
4. There are no references to cited literature in the Results and analysis chapter.
5. Conclusions - are far too long and should be more detailed
6. Text edit notes:
- correct the record of references to the cited literature when the work has 3 authors
- there is no need to write the number in words, e.g. five (5)
- I suggest removing tables 3, 4, 6, 8, 9, 11, 12 - their content is described in the text
- remove the titles in the Figures and the borders of the figures
- codification is unnecessary - it is not used in the work
- correct the References list in accordance with the Sustainability requirements and complete the missing journal titles, page and volume numbers
My biggest complaint is the number of papers this manuscript is based on. I disagree with the authors that there are no works on this topic. I recommend a more thorough search for scientific sources.
Minor editing of English language required
Author Response

(The authors gave the same response as above.)

Round 2
Reviewer 3 Report
The authors have addressed the reviewer's querries. The manuscript is recommended for acceptance.
Author Response
Authors thank Reviewers for the reviews and improvements to the manuscript.
Reviewer 4 Report
The authors have made some changes, however, there are shortcomings in the text of the manuscript that should be corrected.
1. The aim of the work has not yet been clearly defined. It should be placed at the end of the Introduction.
2. I do not understand why the authors use the following notation in the text: twelve (12) articles; should be 12 articles or twelve articles.
3. Conclusions - line 632-639 should be deleted.
Author Response
Point-by-point response to Reviewer 4's comments is attached.
